# Comparison Length of Linker in Compound for Nuclear Medicine Targeting Fibroblast Activation Protein as Molecular Target

**DOI:** 10.3390/ijms252212296

**Published:** 2024-11-15

**Authors:** Kentaro Hisada, Kazuko Kaneda-Nakashima, Yoshifumi Shirakami, Yuichiro Kadonaga, Atsuko Saito, Tadashi Watabe, Sifan Feng, Kazuhiro Ooe, Xiaojie Yin, Hiromitsu Haba, Masashi Murakami, Atsushi Toyoshima, Jens Cardinale, Frederik L. Giesel, Koichi Fukase

**Affiliations:** 1Radiation Biological Chemistry, MS-CORE, Forefront Research Center, Graduate School of Science, Osaka University, 1-1 Machikaneyama-cho, Toyonaka 560-0043, Osaka, Japan; hisadak24@chem.sci.osaka-u.ac.jp (K.H.); fengs24@chem.sci.osaka-u.ac.jp (S.F.); 2Interdisciplinary Research Center for Radiation Sciences, Institute for Radiation Sciences, Osaka University, 2-4 Yamadaoka, Suita 565-0871, Osaka, Japan; yoshifumi_shirakami@irs.osaka-u.ac.jp (Y.S.); kadonagay19@irs.osaka-u.ac.jp (Y.K.); murakami@irs.osaka-u.ac.jp (M.M.); toyo@irs.osaka-u.ac.jp (A.T.); koichi@chem.sci.osaka-u.ac.jp (K.F.); 3Research Center for Ultra-High Voltage Electron Microscopy, Osaka University, 7-1 Mihogaoka, Ibaraki 567-0047, Osaka, Japan; saito@uhvem.osaka-u.ac.jp; 4Department of Radiology, Graduate School of Medicine, Osaka University, 2-2 Yamadaoka, Suita 565-0871, Osaka, Japan; watabe.tadashi.med@osaka-u.ac.jp; 5Radioisotope Research Center, Institute for Radiation Sciences, Osaka University, 2-4 Yamadaoka, Suita 565-0871, Osaka, Japan; ooe@rirc.osaka-u.ac.jp; 6Nishina Center for Accelerator-Based Science Nuclear Chemistry Group, RIKEN, 2-1 Hirosawa, Wako 351-0198, Saitama, Japan; yin.xiaojie.yin@riken.jp (X.Y.); haba@riken.jp (H.H.); 7Nuclear Medicine Department, University Hospital Düsseldorf, Moorenstraße 5 Universitätsklinikum, 40225 Düsseldorf, Germany; jens.cardinale@med.uni-duesseldorf.de (J.C.); frederik.giesel@med.uni-duesseldorf.de (F.L.G.)

**Keywords:** FAPα, At-211, nuclear medicine, middle molecule, FAPI

## Abstract

Novel nuclear medicine therapeutics are being developed by labeling medium-molecular-weight compounds with short-lived alpha-emitting radionuclides. Fibroblast activation protein α (FAPα) is recognized as a highly useful molecular target, and its inhibitor, FAPI, is a compound capable of *theranostics*, both therapeutic and diagnostic, for cancer treatment. In this study, we compared the functions of two compounds that target FAPα: ^211^At-FAPI1 and ^211^At-FAPI2. First, in vitro screening procedures are generally accepted because of the low endogenous expression of FAPα. We suggest the usefulness of this 3D culture system for in vitro screening. Second, when FAPIs are used therapeutically, the expected therapeutic effects are often not achieved. Therefore, we compared the accumulation and excretion in tumor tissues and the anti-tumor effects based on the length of the linker in the compounds. The compounds were rapidly labeled using the *Shirakami reaction*. Doubling the linker length increased tumor retention. Additionally, the excretion pathway was altered, suggesting a potential reduction in toxicity. Although no significant differences were observed in the anti-tumor effects of ^211^At-FAPI1 and ^211^At-FAPI2, it was confirmed that the linker length affects the biological half-life.

## 1. Introduction

Fibroblast activation protein α (FAPα) is a cell membrane protein that belongs to the serine protease family and plays an important role as an enzyme [1,2,3]. It has been reported that FAPα expression in cancer stroma is a factor in the poor prognosis of patients with cancer. Therefore, FAPα is an excellent target for cancer treatment. The development of imaging probes that utilize this target is progressing worldwide, and their usefulness continues [4]. It has been reported that introducing a cyano group (CN, nitride group) into the proline residue of a FAP inhibitor (FAPI) results in an improved FAPI (UAMC-1110) [5]. This became the starting point for discoveries such as FAPI-04 [6] and FAPI-46 [7] by the Heidelberg research team.

We examined the linker type and length based on previous reports [8]. We found no difference in specificity according to linker length and compared polyethylene glycol (PEG) and piperazine (PIP) as linkers. We found that PEG was a better linker, but its accumulation in the lower gastrointestinal tract was a concern. Since FAPI molecularly targets FAPα, a therapeutic effect might not be observed if FAPα is not sufficiently expressed. Since the expression of FAPα in xenograft tumors was slightly higher in the BxPC3 model than that in the previously reported PANC-1 model, we decided to adopt the BxPC3 model in this study. For compound screening, we previously used cells overexpressing FAPα. Other research groups have conducted similar studies; however, because of the artificial nature of these systems [4], the in vitro and in vivo results often differ. Therefore, we examined the screening methods and performance of these compounds. The compounds compared in this study were ^211^At-FAPI1 and ^211^At-FAPI2, the structures of which are shown in Figure 1. The difference between these compounds was the length of the PEG linker. The length of the PEG linker used in this study is selected for the ease of synthesis and comparison. If the linker is longer than ^211^At-FAPI2, the synthesis becomes difficult, and if it is shorter than ^211^At-FAPI1, the linker might become meaningless.

Compounds may behave differently in the body than expected due to minor changes. Because of the enormous expense involved in drug development, more efficient screening methods are required. Various genetically engineered experimental systems are often used to confirm the actual clinical picture. We believe that the gap between cell experiments, animal experiments, and actual clinical effects can be bridged by employing a model that approaches the clinical picture through environmental manipulation rather than genetic manipulation. Therefore, in this study, we simultaneously investigated multiple screening methods. The results of this study will be useful in the development of drugs that target FAPα, regardless of whether they are nuclear medicine therapeutics. The proposed screening method will also help bridge the gap between in vitro and in vivo studies.

## 2. Results

### 2.1. FAPα Expression in 3D Sphere Culture Cells

FAPα is highly expressed in tumor tissues but poorly expressed in monolayer-culture conditions. It is also known that FAPα expression can be induced by co-culturing with CAF, even in monolayer cultures. In other words, FAPα may be induced if the conditions are similar to those of the tumor tissue. Therefore, we employed three-dimensional (3D) sphere culture. If FAPα is upregulated in 3D sphere culture, which has been used for drug screening in recent years, it is expected to be useful as a screening method for FAPα-selective compounds. Figure 2 shows the expression of FAPα in cells in monolayer culture (two-dimensional, 2D) and sphere culture (3D) using PANC-1 cells. The expression levels of FAPα on the cell surface were enhanced under 3D-sphere-culture conditions compared with those under 2D-monolayer-culture conditions. In PANC-1, the intensity of FAPα changed from 2D (*MFI:* 763.33 ± 22.68) to 3D (*MFI:* 1929.33 ± 38.05), and in BxPC3, it changed from 2D (*MFI:* 845.00 ± 3.61) to 3D (*MFI:* 2317.00 ± 37.63) (Figure 2). In this study, cells seeded in different plates at the same density were cultured for 48 h, and 3D sphere formation was visually confirmed before the expression of FAPα on the cell surface was confirmed. The fraction with low FAPα expression remained even under 3D-sphere conditions after 48 h of incubation.

### 2.2. Evaluation of Astatine Labels for FAPI Compounds

Figure 3 shows TLC data for ^211^At-FAPI1 and ^211^At-FAPI2. Although not detailed in this paper, temperature is very important when labeling this compound. In ^211^At-FAPI2, it has been observed that the CN group is converted to CONH_2_ or COOH when reacted at high temperatures. As this phenomenon was observed at processing temperatures above 80 °C, we believe that the processing temperature for labeling should be maintained at approximately 70 °C.

### 2.3. Intracellular Uptake of FAPI1 and FAPI2

We compared the uptake of ^211^At-FAPI1 and ^211^At-FAPI2 by FAPα/HEK293 cells. As no difference was observed in previous reports, we checked the uptake of ^211^At-FAPI1 and ^211^At-FAPI2 earlier (30 min) and later (2 h) than in the previous condition (Figure 4). The uptake of ^211^At-FAPI2 was lower at the earlier time point but increased to almost the same level after 2 h, suggesting a slight difference in the uptake rate of ^211^At-FAPI1 and ^211^At-FAPI2. The results of this study showed a slight difference between the uptake speed of ^211^At-FAPI1 and ^211^At-FAPI2.

Figure 5 compares monolayer (2D) and sphere cultures (3D), confirming that 3D culture increased FAPα expression and uptake. Similar trends were observed in the two cell lines, but the selectivity of the compounds varied by cell line. This is because in PANC-1 cells, ^211^At-FAPI1 uptake increased approximately 10-fold and ^211^At-FAPI2 uptake increased approximately 15-fold, whereas in BxPC3 cells, ^211^At-FAPI1 uptake was 10-fold higher than that in PANC-1 cells, but ^211^At-FAPI2 uptake was approximately 3-fold higher. The difference between 3D and 2D cultures has different values depending on the state (size) of the sphere in 3D culture, but there is no doubt that the uptake is significantly higher than that in 2D culture.

### 2.4. Distribution of Tissues in Tumor Bearing Mice

In previous reports, FAPI compounds were found to accumulate in the gastrointestinal tract. In the present study, we focused on its accumulation in the lower gastrointestinal tract. Although accumulation in the colon, cecum, and small intestine tended to be high for ^211^At-FAPI2, these contents were higher for ^211^At-FAPI1. The length of the linker clearly affected the excretion pathway but did not significantly affect its distribution in other organs (Table 1).

### 2.5. Distribution in BxPC3 Tumors

We compared the distribution of ^211^At-FAPI1 and ^211^At-FAPI2 in a BxPC3 tumor. The reason for using BxPC3 instead of PANC-1 is that it has been reported that BxPC3 tumors have better CAF development in xenograft models [11] than ^211^At-FAPI1 and ^211^At-FAPI2 accumulation. We compared the results 3 and 24 h after administration in % ID/g of BxPC3 tumors. Although there was a tendency for accumulation to be higher in ^211^At-FAPI2 than in ^211^At-FAPI1, the difference was not statistically significant (Figure 6).

### 2.6. Excretion of FAPI1 and FAPI2 from Animals

Accumulation in the tumor tissue alone tended to be higher for ^211^At-FAPI2, although the difference was not significant. The organs in which significant differences were found were the kidneys, the liver, and the gastrointestinal tract. In terms of the total amount excreted from the body, ^211^At-FAPI2 was less likely to be excreted, with the greatest difference observed in urinary excretion via the kidneys (Figure 7).

### 2.7. Anti-Tumor Effects of ^211^At-FAPI1 and ^211^At-FAPI2

The anti-tumor effects of ^211^At-FAPI1 and ^211^At-FAPI2 are shown in Figure 8. Tumor growth inhibition was observed relatively early after drug administration (Figure 8A), with no significant difference observed between ^211^At-FAPI1 and ^211^At-FAPI2. Significant inhibition of tumor growth was observed in each group. At approximately one month, there was an increase in tumor growth in all groups, possibly due to repopulation. There were significant differences in tumor wet weight between the control and ^211^At-FAPI1 groups and between the control and ^211^At-FAPI2 groups, but not between the ^211^At-FAPI1 and ^211^At-FAPI2 groups (Figure 8B). Weight loss was lower in the ^211^At-FAPI 1 and ^211^At-FAPI2 groups than that in the control group (Figure 8C).

## 3. Discussion

As FAPα is expressed in stromal cells, it is not present on the cell membrane surface under normal culture conditions. We used 3D sphere culture. It has been reported that 3D sphere cultures alter the expression of surface molecules (particularly adhesive molecules) and drug resistance in multiple pancreatic cancer cell lines. In Hwang’s research, by co-culturing with stellate cells, the conditions were brought closer to in vivo conditions, but changes were also observed, even with 3D sphere culture alone [12]. In 3D sphere culture, cells are no longer present in a single layer and can resemble a tumor. Researchers have long used 2D monolayer cultures to study cell and disease mechanisms. Two-dimensional monolayer cell culture models have been a convenient and cost-effective method for culturing and conducting experiments; however, in recent years, 3D sphere culture has been used to better reproduce in vivo tissues because of its physiological relevance. In living organisms, cells do not grow as a monolayer without the inclusion of other cells or tissues, and most cells exist as complex 3D structures composed of different cell types within the extracellular matrix. Numerous cell–cell and cell–matrix interactions influence various properties. Additionally, 2D monolayer cultures provide uniform access to nutrients and oxygen, which is not the case for cell clusters such as cancer cells. Three-dimensional cancer spheroids reflect cancer in vivo more closely, with inner cells having less access to nutrients and oxygen than cells in the outer layer, thus creating a natural gradient. The results of the comparison of cellular uptake between 2D and 3D cultures of cancer cells showed that, unlike the data from the artificial overexpression system, there were large differences between cell lines. If the screening of compounds was only for selectivity, there would be no problems with the overexpression system. However, if one is looking for compounds that are more useful in specific cancer types, 3D culture may be a useful tool, as its uptake is higher than that in monolayer culture. In addition, because it can be performed simply by changing the culture plates, there is no need for genetic recombination experiments, and it is very simple to perform.

The FAPα overexpressing cell line used in this experiment was re-cloned and used as a cell line with a higher FAPα expression than that used in the previous report [8]. This is because a decrease in FAPα was observed after long-term culture. The expression level of FAPα might also affect the proliferation of cancer cells themselves. Because we confirmed that the growth speed is significantly affected when FAPα-overexpression lines are constructed, especially in breast cancer cells. The effect was particularly pronounced when subcutaneous tumors were constructed. This might be one of the facts that makes the development of FAPα molecularly targeted drugs difficult.

It has been reported that ^211^At-FAPI1 (PEG linker) and ^211^At-FAPI5 (PIP linker) tend to accumulate in the gastrointestinal tract [8]. To reduce the accumulation of ^211^At-FAPI compounds in the gastrointestinal tract, we increased the linker length. In previous studies, we confirmed that increasing the length of the linker did not affect the selectivity; therefore, we attempted to verify this in animals [8]. By changing the length of the linker, the physical properties could be changed without a significant change in selectivity. As a result, ^211^At-FAPI2 increased its internal retention compared to ^211^At-FAPI1, but did not increase tumor specificity (Table 1, Figure 6 and Figure 7). This was because the excretion route changed from being urinary to gastrointestinal. In other words, it was not necessary to lengthen the linker for this compound. There have also been reports on the design of FAPI as polymers. Since 2021, there have been several reports of dimerizing FAPI to increase the specificity of the compound [13,14,15,16,17,18,19,20,21,22,23]. There are no reports on trimers, and only one tetramer has been reported [14]. The number of recognition sites may indeed increase specificity; it is extremely difficult to synthesize and becomes unwieldy as the molecular weight increases.

The length of the linker and its effect on uptake has also been reported for other compounds [24,25]. The results of this study suggest that linker length may increase tissue uptake and at the same time inhibit efflux from the body. However, no significant differences in therapeutic efficacy were observed when testing the therapeutic effect in animal models. However, controlling efflux can improve patient safety. Controlling the water solubility and fat solubility by varying the length of the linker could be useful when designing compounds.

As we develop nuclear medicine therapeutics, we are examining the structure to take on the behavior we desire. However, sometimes it goes the way we want it to, and sometimes it does not. The position of the side chain under consideration is also important. It might be not only the distribution of compounds but also the side chain of the compounds nearby that may be affected by the alpha rays emitted by the labeled radionuclide [26].

## 4. Materials and Methods

### 4.1. Materials

The designed compound was manufactured by the Peptide Research [8] Institute (Ibaraki, Osaka, Japan). For the detailed structure of precursors and synthetic procedures, please refer to previous reports [8]. The structure of this compound is shown in Figure 1.

### 4.2. Manufacturing of Astatine-211(^211^At) and Labeling

We manufactured ^211^At based on existing methods [10]. After producing ^211^At, dry separation was performed, and the resulting ^211^At aqueous solution was used to label the compound. The compound was labeled using the “*Shirakami reaction*” as previously reported [9]. The labeling efficiency of ^211^At-FAPI1 and ^211^At-FAPI2 and quality were evaluated by way of TLC [8].

### 4.3. In Vitro Evaluation

#### 4.3.1. Cell Culture

We used the BxPC3 and PANC-1 human pancreatic cancer cell lines. The cells were obtained from the American Type Culture Collection (ATCC). The cells were maintained according to the manufacturer’s instructions. All the culture media were purchased from Fujifilm Wako Pure Chemical Industries (Osaka, Osaka, Japan). We also used the FAPα-overexpressing HEK293 cell line (FAPα/HEK293). These cell lines have been used in past research [8]. The parental HEK293 cell line was used as a negative control. The cells were maintained in E-MEM supplemented with 10% heat-inactivated fetal bovine serum (Thermo Fisher Scientific), 1% non-essential amino acids, and 1% penicillin-streptomycin (Fujifilm Wako Pure Chemical). For the 3D sphere culture, a special round bottom plate (spheroid microplate) was used (Thermo Fisher Scientific).

#### 4.3.2. Evaluation of FAPα Expression

Because FAPα expression on the cell surface is low in monolayer cultures, overexpression systems are often used [4]. However, since the expression level increases in xenograft cancer tissues, it was thought that the expression level of FAPα would change when cells formed spheres, so we confirmed this. For surface staining, the cells were stained with FAPα antibody (R&D Systems, Minneapolis, MN, USA) at room temperature for 15 min, washed with PBS (-), and then resuspended in MACS buffer (PBS, pH 7.2, with 0.5% bovine serum albumin, and 2 mM EDTA). The cells were washed once with PBS and resuspended in MACS buffer before analysis on an Attune NxT Flow cytometer (Thermo Fisher Scientific).

#### 4.3.3. Uptake Assay Under 3D Culture Condition

PANC-1 and BxPC3 cells were seeded in a spheroid microplate (Thermo Fisher Scientific). FAPα expression is very low in monolayer cultures but increases in tumor tissues. Therefore, we adopted spheroid culture and conducted experiments under conditions that mimicked those in which FAPα was expressed. After incorporating the labeled compound as in our previous reports, the cells were washed and then lysed with 0.1 M NaOH (Fujifilm Wako Pure Chemical), and the amount of labeled compound incorporated into the cells was measured using a γ-counter (Wizard^2^ 2480, Perkin Elmer, MA, USA). Radiation intensities were corrected by cell number to measure protein levels. The protein content was measured using a BCA protein assay kit (Fujifilm Wako Pure Chemical). Monolayer culture (2D) uptake experiments were performed as previously reported.

### 4.4. In Vivo Experiment

#### 4.4.1. Distribution of Tissues

Five-week-old Balb/c-nu/nu mice were purchased from Japan SLC (Shizuoka, Japan). The mice were used after 1 week of habituation. The cells were grown as described in the in vitro experiments. The cell suspension was adjusted to 1 × 10^8^ cells/mL, mixed 1:1 with Matrigel, and transplanted into the backs of the mice. The cells (1 × 10^7^) were transplanted into each mouse. Two weeks later, tumor engraftment was confirmed before experiments were performed. After preparing the ^211^At labeled FAP1 and 2, it was divided into a syringe, and the dose was measured using an ICG-8 Curie meter (ALOKA, Tokyo, Japan). ± two groups for ^211^At-FAPI1 (N = 6, body weight: 22.44 ± 0.29 g) and ^211^At-FAPI2 (N = 6, body weight: 21.78 ± 0.70 g). Half of the specimens were dissected after 3 h, and the rest were dissected after 24 h. After administration through the tail vein of the mice, the syringe was re-measured to calculate the exact dose. The tumors were dissected one hour after administration, the mice were dissected, and the amount of uptake in the tissues was calculated.

#### 4.4.2. Comparison of Anti-Tumor Effect Between ^211^At-FAPI1 and ^211^At-FAPI2

Labeled compounds were prepared as in the distribution experiment, and the mice were divided into control (body weight: 18.68 ± 0.58 g), ^211^At-FAPI1 (body weight: 18.46 ± 0.61 g), and ^211^At-FAPI2 (body weight: 18.58 ± 0.35 g) groups. Each treatment group consisted of 5 mice. ^211^At-FAPI1 and ^211^At-FAPI2 were administered, respectively: ^211^At-FAPI1 (712.54 ± 26.20 kBq/mouse) and ^211^At-FAPI2 (731.80 ± 33.04 kBq/mouse). Saline was used as the control. Observations were made after administration, and tumor size and weight were measured three times per week. Tumor volume was approximated by the following formula: V = (W^2^ × L)/2 (V: tumor volume, W: minor axis, width, L: major axis, length). The endpoint was defined as a tumor size exceeding 10% of the body weight or the point at which the health of the control group was deemed to have deteriorated significantly, even if the size did not reach 10%. The wet weights of the excised tumors were measured at the end of the observation period.

### 4.5. Confirmation by Chemical Experiment

#### TLC Analysis

Labeled compounds treated under each condition were spotted onto a TLC membrane (Merck KGaA, Darmstadt, Germany) and developed using a developing solvent (Water: A acetonitrile = 1:2). Detection was performed using an imaging plate with an FLA7000 (Cytiva, Tokyo, Japan).

### 4.6. Statistical Analysis

The results are expressed as the mean ± standard error. Comparisons between the groups were performed using unpaired *t*-tests in Microsoft Excel (version 2016). For multiple comparisons among the three groups, a Bonferroni correction was performed. Differences were considered statistically significant at *p* < 0.05.

## 5. Conclusions

It has been reported that the length of the PEG linker affects its retention in tumors, even if the drugs are different [25]. By extending the linker, it was possible to promote excretion without reducing tumor tissue accumulation or anti-tumor effects. Another proposal is the screening method. It was difficult to reproduce FAPα expression in a 2D monolayer culture, but by employing a 3D sphere culture, FAPα expression could be induced without genetic modification. This method is very simple and requires only a change in the culture plate type.

## Figures and Tables

**Figure 1 ijms-25-12296-f001:**
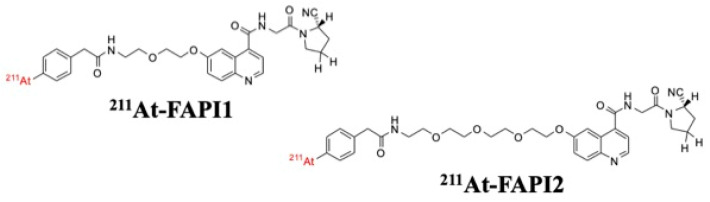
Structures of ^211^At-FAPI1 and ^211^At-FAPI2. The only difference between these compounds was the length of the linker. In ^211^At-FAPI2, the number of PEG molecules was twice that in ^211^At-FAPI1. These were rapidly labeled by the *Shirakami reaction* [9] based on previously reported precursors [8] using ^211^At produced based on previous reports [10].

**Figure 2 ijms-25-12296-f002:**
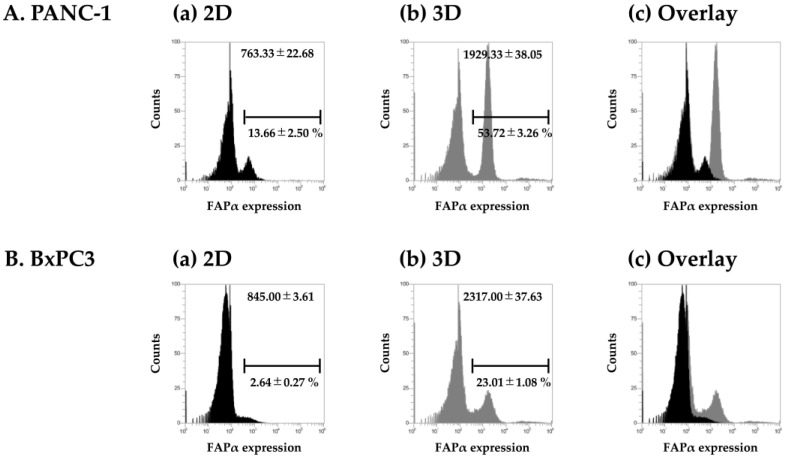
Comparison between normal- and sphere-culture conditions. FAPα expression levels in PANC-1 and BxPC3 cells, (**A**) PANC-1 and (**B**) BxPC3. (**a**) Two-dimensional culture (monolayer) and (**b**) three-dimensional culture (sphere). (**c**) Overlay of (**a**) and (**b**). % of FAPα-positive cells and mean flow intensities are shown in the figure. Human Fibroblast Activation Protein alpha/FAP-APC-conjugated antibody (R&D systems) #FAB3715A-025. Cell analyzer: Attune NxT (Thermo Fisher Scientific Inc., Waltham, MA, USA). Cells were cultured in a sphere low-attachment surface plate (Thermo Fisher Scientific). Mean fluorescence intensities (MFIs) were obtained in triplicate, and the mean value of MFI was shown in the histogram. Values presented at the top of the histogram indicate MFI, and values presented at the bottom indicate the percentage of FAPα-positive cells to the total cell count.

**Figure 3 ijms-25-12296-f003:**
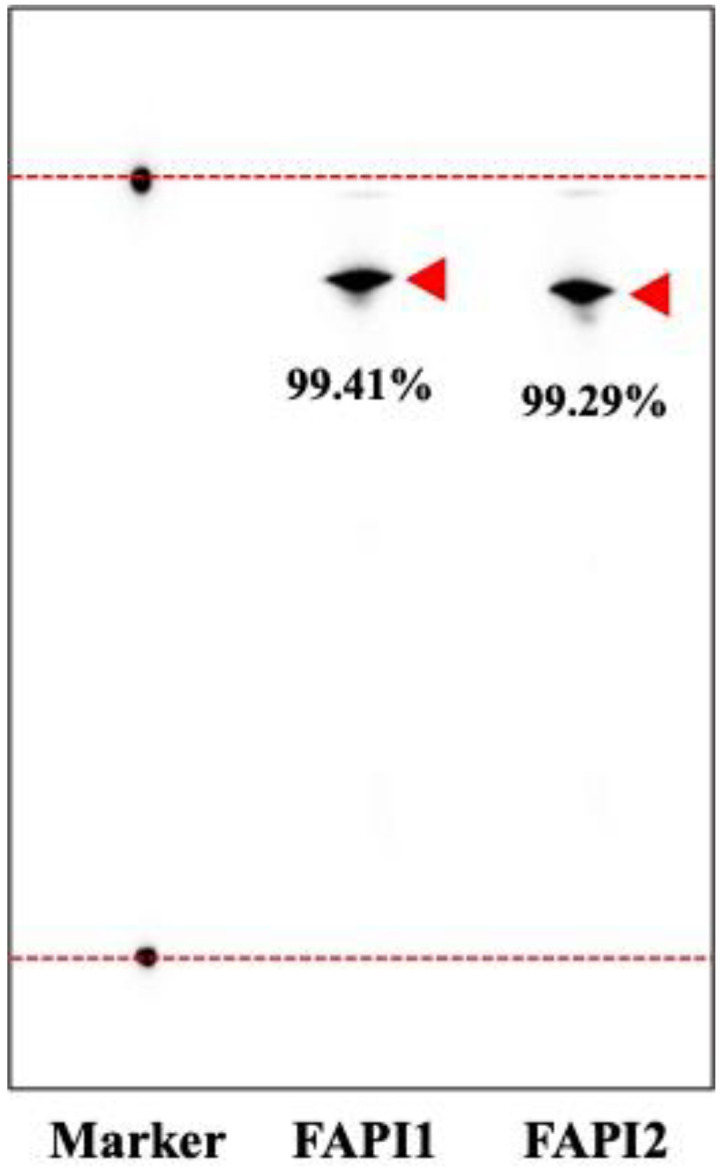
Confirmation of ^211^At-FAPI1 and ^211^At-FAPI2 using TLC analysis. The astatine label of the FAPI compound was evaluated by TLC. FAPI1 indicates ^211^At-FAPI, and FAPI2 indicates ^211^At-FAPI2.

**Figure 4 ijms-25-12296-f004:**
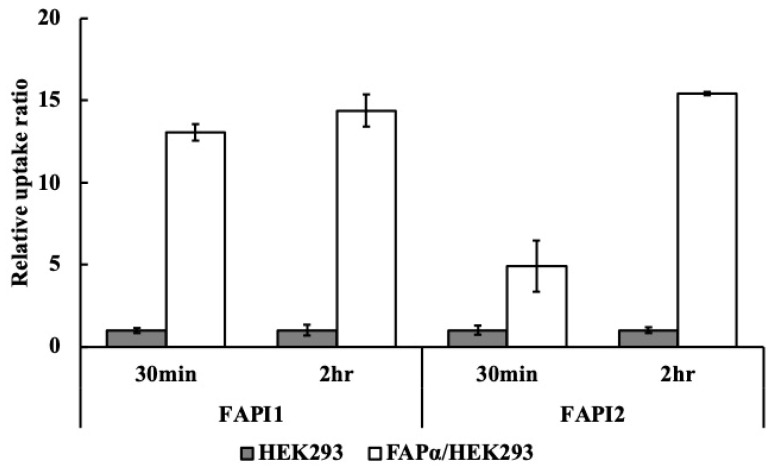
Cellular uptake of FAPI1 and FAPI2 depends on FAPα expression-level. ^211^At-FAPI1 and ^211^At-FAPI2 cells were treated with HEK293 or FAPα/HEK293 cells. The cells were collected after 30 min or 2 h. The *Y*-axis was calculated, and the values of %ID/μg protein of FAPα/HEK293 were divided by those of HEK293 cells. FAPI1 indicates ^211^At-FAPI and FAPI2 indicates ^211^At-FAPI2. The difference between HEK293 and FAPα/HEK293 cells was significant at *p* < 0.05.

**Figure 5 ijms-25-12296-f005:**
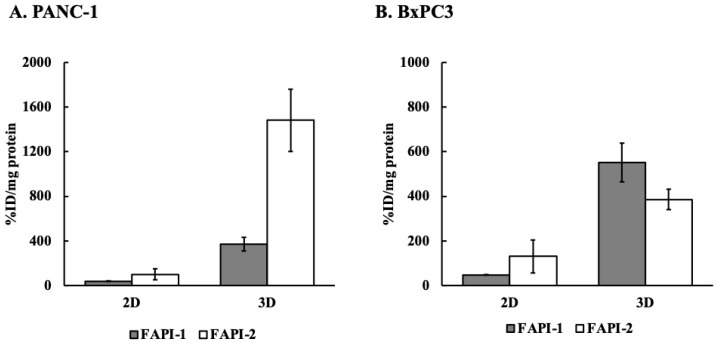
Comparison of intracellular uptake between 2D and 3D cultures of (**A**) PANC-1 and (**B**) BxPC3 cells. The cells were then treated with ^211^At-FAPI1 or FAPI2. Gray histograms indicate ^211^At-FAPI1 uptake, and white histograms indicate ^211^At-FAPI2 uptake. Two-dimensional culture shows the uptake under monolayer-culture conditions, and three-dimensional culture shows the uptake under sphere-formed conditions.

**Figure 6 ijms-25-12296-f006:**
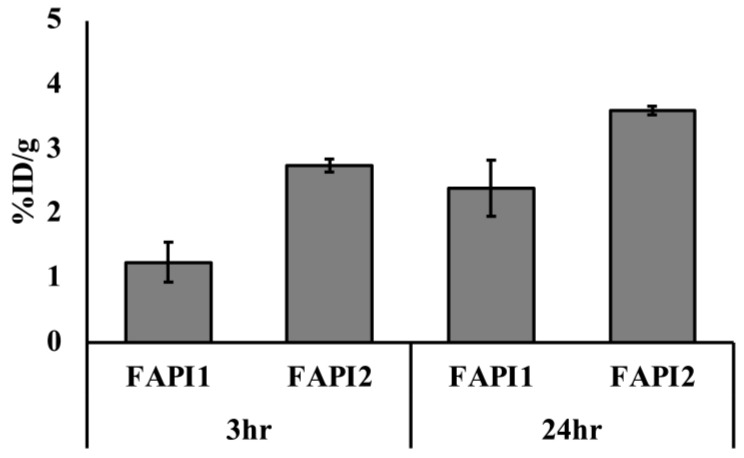
%ID/g in tumor in vivo. ^211^At-FAPI1 and ^211^At-FAPI2 were administered to the BxPC3 tumor model and dissected 3 h (3 h) or 24 h (24 h) after administration. FAPI1 indicates ^211^At-FAPI and FAPI2 indicates ^211^At-FAPI2. No significant differences were observed between FAPI1 and FAPI2 expression.

**Figure 7 ijms-25-12296-f007:**
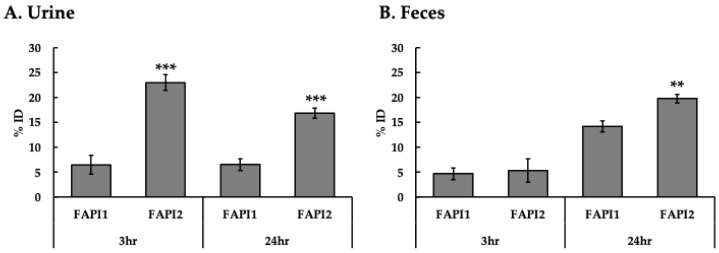
Amount of excretion in vivo. ^211^At-FAPI1 and ^211^At-FAPI2 were administered to the BxPC3 tumor model and dissected 3 or 24 h after administration. (**A**) % ID in urine. (**B**) % ID in feces. There were significant differences between FAPI1 and FAPI2. ** *p* < 0.01, *** *p* < 0.001.

**Figure 8 ijms-25-12296-f008:**
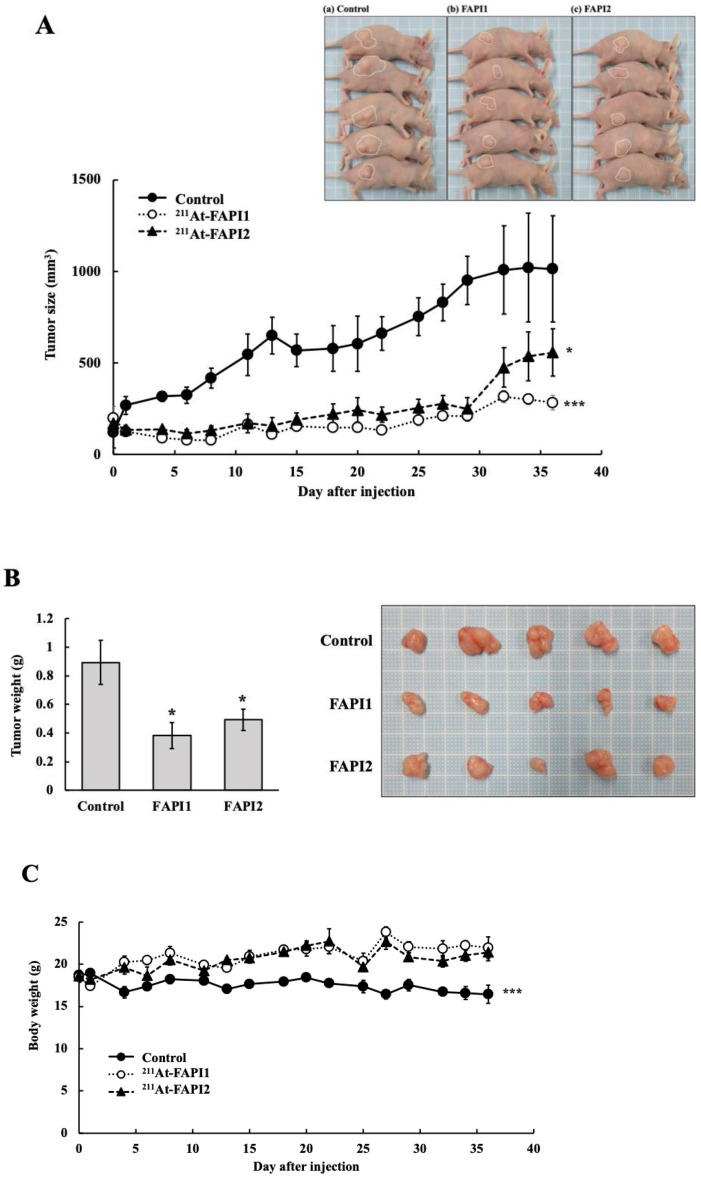
Anti-tumor effects of FAPI1 and FAPI2. (**A**) is the tumor size in ^the 211^At-FAPI1 and ^211^At-FAPI2 groups. (**B**) Body weights of the experimental mice. (**C**) Tumor weights from xenograft mice. * *p* < 0.05, *** *p* < 0.001.

**Table 1 ijms-25-12296-t001:** Distribution of tissues.

	3	hr	24	hr
(%ID/g)	FAPI1	FAPI2	FAPI1	FAPI2
**Tumor**	1.24 ± 0.43	2.75 ± 0.35	2.39 ± 0.87	3.61 ± 0.63
**Gall bladder**	17.54 ± 16.23	38.05 ± 9.39	2.04 ± 0.65	3.23 ± 2.35
**Liver**	0.99 ± 0.10	1.67 ± 0.17 *	0.43 ± 0.02	0.93 ± 0.13 *
**Kidney**	1.66 ± 0.13	2.88 ± 0.18 **	1.09 ± 0.04	1.96 ± 0.11 **
**Colon**	2.66 ± 1.19	5.05 ± 1.19	0.87 ± 0.07	1.58 ± 0.07 **
**Colon contents**	11.12 ± 42.88	51.77 ± 20.60	3.98 ± 1.04	21.62 ± 9.44
**Small intestine**	3.05 ± 0.77	4.08 ± 0.31	0.80 ± 0.04	3.85 ± 2.24
**Small intestine contents**	25.50 ± 9.34	14.56 ± 3.14	1.44 ± 0.29	6.54 ± 3.76
**Cecum**	7.43 ± 0.41	13.79 ± 1.91 *	1.18 ± 0.07	4.23 ± 2.09
**Cecum contents**	53.13 ± 8.64	50.06 ± 7.52	3.02 ± 0.61	11.08 ± 4.19

All groups consisted of three mice. The mice were euthanized and dissected 3 and 24 h after the administration of the labeled compounds. FAPI1 indicates ^211^At-FAPI and FAPI2 indicates ^211^At-FAPI2. * *p* < 0.05, ** *p* < 0.01.

## Data Availability

The original contributions presented in this study are included in the article. Further inquiries can be directed to the corresponding author.

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
