# Peer review of "Comparison Length of Linker in Compound for Nuclear Medicine Targeting Fibroblast Activation Protein as Molecular Target"

_ijms, 2024, doi:10.3390/ijms252212296_

Round 1
Reviewer 1 Report
Comments and Suggestions for Authors
I would like to thank the authors for this interesting and well-structured study on Astat-labelled FAP compounds.
I recommend acceptance in the current form.
Author Response
Dear Reviewer 1,
We sincerely appreciate your favorable comments.
Your encouraging feedback is greatly valued.
We have revised our article based on the comments provided by other reviewers.
We aspire to enhance the quality of our article.
Thank you.
Kazuko Kaneda-Nakashima, Ph.D.
MS-CORE, FRC, Osaka University Graduate School of Science
Reviewer 2 Report
Comments and Suggestions for Authors
Comparison length of linker in compound for nuclear medicine 2 targeting FAP as a molecular target by Kentaro Hisada et al. is a description of two FAPα-targeting compounds differing only by the length of otherwise identical PEG linkers. The longer linker affected washout but not targeted binding, potentially reducing total absorbed dose through shunting clearance from urinary to fecal. This manuscript is well written and clear except for a few discrepancies listed below:
Line 41: an enzymatic enzyme
Line 44: change “property” to “target” or “biomarker”
Line 45: eliminate “without saying”.
Line 47: “an improved” over “an excellent”
Line 68: “Was” should be “is”
FAPα is most strongly expressed on stellate cells and CAFs in the stroma. Why did the authors try to conduct binding on epithial cells in a sphere? In clinic, it will be stromal cell binding taking the most of available radioligand.
Lines 94-95: Please include units for all values in the text.
Line 226: By forcing FAPα expression on pancreatic cancer subculture, you may be getting a different glycosylation pattern or nitrosylation, altering the binding of your radioconjugates. You should probably do a Ki measurement again to make sure you have a representative protein.
Line 288. Does this antibody recognize both overexpressed epithelial FAPα and stromal FAP expression?
Line 328: could you please provide the equation used to calculate the tumor mass?
Line 352: Please erase the comment there and indicated whether the authors have related patents.
Author Response
Dear Reviewer 2
Thank you very much for taking the time to review this manuscript. Please find the detailed responses below and the corresponding revisions/corrections highlighted/in track changes in the re-submitted files.
Point-by-point response to Comments:
Comments 1: Line 41: an enzymatic enzyme
Response 1: Thank you for pointing this out. We agree with this comment. Therefore, we rewrite this point “an enzyme”.
Comments 2: Line 44: change “property” to “target” or “biomarker”
Response 2: Agree. We changed “property” to “target”.
Comments 3: Line 45: eliminate “without saying”.
Response 3: Agree. We eliminated “without saying”.
Comments 4: Line 47: “an improved” over “an excellent”
Response 4: Agree. We turned “an excellent” to “an improved”.
Comments 5: Line 68: “Was” should be “is”
Response 5: Agree. We turned “was” to “is”.
Comments 6: FAPα is most strongly expressed on stellate cells and CAFs in the stroma. Why did the authors try to conduct binding on epithelial cells in a sphere? In clinic, it will be stromal cell binding taking the most of available radioligand.
Response 6: Thank you for your comments. Since FAPα is expressed in vivo in tumor tissues, we wondered if there were conditions under which FAPα could be overexpressed without genetic modification. We hypothesized that there might be conditions under which FAPα could be overexpressed without genetic recombination. We found that FAPα expression was upregulated in the cell lines we used in our carcinoma model when cultured in the absence of scaffolds. Although the regulatory mechanism of FAPα expression remains to be verified, our results suggest that even cell lines that are supposed to be single clones change their properties when cultured in heterogeneous conditions. Therefore, we performed the experiments described in the text to investigate the possibility of using the FAPα highly selective compounds.
Comments 7: Lines 94-95: Please include units for all values in the text.
Response 7: Thank you for your comments. These values are the mean fluorescent intensity. We have indicated that it is the value of MFI in the text and added an explanation in the footnote of the Figure 2.
Comments 8: Line 226: By forcing FAPα expression on pancreatic cancer subculture, you may be getting a different glycosylation pattern or nitrosylation, altering the binding of your radioconjugates.
Response 8: Thank you for your comments. Our compounds are used as couriers for delivery of radioisotopes to cancer tissue. The amount used is in microdoses and the toxicity of the compound itself is almost negligible. No toxic metabolites have also been detected. Changes in the compound after achieving delivery to cancer tissue are not a problem for this application.
Comments 9: You should probably do a Ki measurement again to make sure you have a representative protein.
Response 9: Thanks a lot for your advice. To evaluate the binding ability of this compound, it is indeed desirable to measure the Ki value. We are currently preparing to set up a system to calculate Ki values for compounds that molecularly target FAPα, but since we have not found an appropriate inhibitor, we did not obtain Ki values, but only compared the amount of uptake. In the future, after selecting an inhibitor, we would like to obtain Ki values in the overexpression system using HEK293 and compare the binding ability of the compounds in a unified manner.
Comments 10: Line 288. Does this antibody recognize both overexpressed epithelial FAPα and stromal FAP expression?
Response 10: Thank you for your comments. The antibodies used in this study were designed specifically for flow cytometer applications, so while they can detect FAPα expressed on the cell surface, they are not suitable for the detection of FAPα expressed in tissue. If immunohistochemical staining is to be performed, we believe that an antibody for a different epitope should be used.
Comments 11: Line 328: could you please provide the equation used to calculate the tumor mass?
Response 11: Thank you for your comments. Tumor volume was approximated by the following formula.
V=(W2 × L) / 2
(V: tumor volume, W: minor axis, width, L:major axis, length)
Comments 12: Line 352: Please erase the comment there and indicated whether the authors have related patents.
Response 12: Thank you for pointing this out. Authors have related patents however those patents were related with not only this article but also another articles. Thus, we erase this section.
Many thanks for your many kind and meaningful comments. We have answered your comments to the best of our ability. The revisions are also highlighted in the text. Thank you in advance.
Kazuko Kaneda-Nakashima, Ph.D.
MS-CORE, FRC, Osaka University Graduate School of Science
Reviewer 3 Report
Comments and Suggestions for Authors
It is a very interesting and well-written manuscript.
The study is well-positioned within the evolving field of nuclear medicine and oncology, focusing on the fibroblast activation protein (FAP) as a molecular target for theranostic applications. The comparison of linker lengths in 211At-FAPI1 and 211At-FAPI2 provides valuable insights into how structural modifications influence compound retention and excretion, a relevant topic for optimizing radiopharmaceuticals.
I only have a few questions/comments:
- The paper mentions that high temperatures impact labeling efficiency for 211At-FAPI2, yet does not provide a detailed rationale for choosing 70°C as the optimal temperature. More extensive temperature studies or a brief justification would make this choice more convincing.
- The study could benefit from a control comparison between the PEG and PIP linkers as explored in previous studies. This would contextualize the impact of PEG linker length on tissue uptake and excretion pathways.
- The paper effectively employs both 2D monolayer and 3D sphere cultures to evaluate FAPα expression, which is an essential aspect given the known limitations of 2D cultures in replicating in vivo conditions. The use of 3D culture in this context is commendable, as it more accurately mimics the tumor microenvironment and has implications for improving drug screening models: Additional information on the characteristics of these 3D cultures, such as the size and morphology of the spheroids or how these parameters might affect FAPα expression, could enrich the study. Also, a more detailed comparison of drug uptake and retention in these different culture conditions would strengthen the argument for 3D models in future FAPα-targeted research.
Author Response
Dear Reviewer 3,
Thank you very much for taking the time to review this manuscript. Please find the detailed responses below and the corresponding revisions/corrections highlighted/in track changes in the re-submitted files.
Point-by-point response to Comments:
Comments 1: - The paper mentions that high temperatures impact labeling efficiency for 211At-FAPI2, yet does not provide a detailed rationale for choosing 70°C as the optimal temperature. More extensive temperature studies or a brief justification would make this choice more convincing.
Response 1: Thank you for pointing this out. In this regard, Dr. Shirakami, who is also the discoverer of the reaction, has examined the labeling efficiency and purity by shaking the temperature and the amount of carrier in detail. The optimal temperature for labeling depends on the structure of the compound. For amino acid derivatives, for example, 50°C is the optimum temperature. If the temperature is too high, the stability of the compound itself will deteriorate. Reactivity improves as temperature is increased up to a certain point, but then it plateaus. The relationship between temperature and reaction is very interesting, and we are currently examining the correlation between the reaction conditions and the structure of the compound.
Comments 2: The study could benefit from a control comparison between the PEG and PIP linkers as explored in previous studies. This would contextualize the impact of PEG linker length on tissue uptake and excretion pathways.
Response 2: Thank you very much for your kind comment. In this study, we were able to show that even minor modifications to a compound can significantly change its effect on the organism. We believe that this knowledge will be useful for future drug development.
Comments 3: The paper effectively employs both 2D monolayer and 3D sphere cultures to evaluate FAPα expression, which is an essential aspect given the known limitations of 2D cultures in replicating in vivo conditions. The use of 3D culture in this context is commendable, as it more accurately mimics the tumor microenvironment and has implications for improving drug screening models: Additional information on the characteristics of these 3D cultures, such as the size and morphology of the spheroids or how these parameters might affect FAPα expression, could enrich the study. Also, a more detailed comparison of drug uptake and retention in these different culture conditions would strengthen the argument for 3D models in future FAPα-targeted research.
Response 3: Thank you very much for your kind comment. Introducing FAPα into cancer cells often resulted in trait changes in our laboratory. Therefore, we were searching for an experimental system that could somehow induce spontaneous expression of FAPα.
We were lucky to find a relatively simple way to enhance expression without using feeder cells or special chambers. We have also observed this phenomenon in some breast cancer cells, and we intend to examine the relationship between 3D sphere size and expression, as well as the relationship with other markers such as SMA and Integrin α3β1, an adhesion molecule associated with FAPα, in the future.
Many thanks for your many kind and meaningful comments. We have answered your comments to the best of our ability. Thank you in advance.
Kazuko Kaneda-Nakashima, Ph.D.
MS-CORE, FRC, Osaka University Graduate School of Science